# Mechanism of One-Step Hydrothermally Synthesized Titanate Catalysts for Ozonation

**DOI:** 10.3390/molecules27092706

**Published:** 2022-04-22

**Authors:** Geshan Zhang, Anhua Jiang, Xinwen Huang, Tian Yuan, Hanrui Wu, Lichun Li, Zongjian Liu

**Affiliations:** 1College of Chemical Engineering, Zhejiang University of Technology, Hangzhou 310014, China; 2111901017@zjut.edu.cn (T.Y.); 2112001011@zjut.edu.cn (H.W.); lichunli@zjut.edu.cn (L.L.); zjliu@zjut.edu.cn (Z.L.); 2College of Environment, Zhejiang University of Technology, Hangzhou 310014, China; 1111827011@zjut.edu.cn

**Keywords:** catalytic ozonation, titanate nanotube, active oxidizing species, phenol, water treatment

## Abstract

A titanate nanotube catalyst for ozonation was synthesized with a simple one-step NaOH hydrothermal treatment without energy-consuming calcination. The synthesized titania catalysts were characterized by X-ray diffraction (XRD), Raman, porosimetry analysis, high-resolution transmission electron microscopy (HR-TEM), Fourier transformed infrared (FTIR), and electron paramagnetic resonance (EPR) analysis. The catalyst treated with a higher concentration of NaOH was found to be more catalytically active for phenol removal due to its higher titanate content that would facilitate more OH groups on its surface. Furthermore, the main active oxidizing species of the catalytic ozonation process were recognized as singlet oxygen and superoxide radical, while the hydroxyl radical may only play a minor role. This work provides further support for the correlation between the properties of titania and catalytic performance, which is significant for understanding the mechanism of catalytic ozonation with titania-based materials.

## 1. Introduction

Ozonation is one of the most widely applied technologies in drinking and waste water treatment plants for removing contaminants. However, it has unavoidable disadvantages, including a relatively high oxidizing selectivity and a high pH dependency, which limit its environmental application. For instance, the drinking water treatment plant revealed a cyano-HABs incident in 2013 in Lake Erie in Toledo (US) even with the ozone treatment [1]. Therefore, it is necessary to apply more advanced technologies to improve the effectiveness of the treatment [2,3,4].

A more economical approach is the modification of the existing ozonation process, i.e., catalytic ozonation [2,5,6]. This technology includes homogeneous catalytic ozonation and heterogeneous catalytic ozonation usually involving transition metals, while the latter would be more practical in water treatment plants due to its heterogeneity and relative immobility that would facilitate the subsequent treatment. Among these solid metals or metal oxide catalysts, many researchers believe TiO_2_ is a type of effective catalyst that can enhance the oxidizing efficiency of the ozonation process. Dr. Song et al. stated that rutile TiO_2_ with a higher specific surface area and more surface OH groups would have a higher catalytic activity for phenol removal [7]. Dr. Ma’s research group synthesized several TiO_2_ catalysts, including nano-TiO_2_ [8], TiO_2_/Silica-Gel [9] and TiO_2_/Zeolite [10], which showed various catalytic behaviors for removing organic pollutants. Anandan et al. studied the degradation of fipronil with P25 TiO_2_ catalytic ozonation treatment and achieved an improvement in efficiency of 15% compared to the ozone-only process [11]. The application of TiO_2_ catalysts in ozonation would accelerate the generation of reactive radical species (such as hydroxyl radical, singlet oxygen, and superoxide radical) in the system, which could result in a lower reaction selectivity and a higher reaction efficiency of ozonation for pollutant removal in water [1,9,12].

TiO_2_ (or titanate) nanotubes have been studied for their chemical, synthetic and environmental application [13,14,15]. Kitano et al. synthesized a titanate nanotube derived from the scrolling of lamellar titanate nanosheets for benzyltoluene formation, which showed a higher catalytic activity due to its lattice distortion and mesoporous structure [16]. Titanate nanotubes have been studied for their adsorption and photocatalysis properties, showing that nanotubes with a lower Na content would have lower adsorption capacity but higher mineralization ability, while nanotubes calcined at 400–500 °C would promote photocatalytic activity [17,18]. In addition, titanate nanotubes were applied to improve the efficiency of ozonation. For instance, Liu et al. examined the effect of calcination temperature on the activity of TiO_2_ catalytic ozonation and found lower annealing temperature would result in better catalytic activity for ammonia removal [19]; Xing et al. also found that titanate nanotubes could accelerate the degradation of *p*-nitrophenol in the ozonation process [20]. To the best of the authors’ knowledge, all the applied titanate nanotubes were calcined before use, many of which were modified based on the reference [21]. In our preliminary experiment, titanate nanotubes without calcination were also found to be catalytically active, which is more efficient in energy utilization and hence attracted our research interest. Moreover, the correlation between the properties of nanotubes and the catalytic performance remains unclear, which is valuable for understanding the mechanism of catalytic ozonation with titania-based materials.

In this study, titania catalysts were synthesized using a simple hydrothermal process without calcination. The catalytic activities of these catalysts were evaluated for phenol (one of the most common and typical organic pollutants in various waters) removal during the ozonation process. The catalysts were characterized by several techniques and investigated for their essential properties responsible for their catalytic activity. Moreover, the main active oxidation species of catalytic ozonation were studied to unveil the catalytic mechanism.

## 2. Results and Discussion

### 2.1. Catalytic Activities of Titania Catalysts

The synthesized titania catalysts were tested for their catalytic activity during the ozonation of phenol. Figure 1 exhibits the concentration changes in phenol through ozonation catalyzed by TiO_2_ (anatase or rutile) treated with different concentrations of NaOH in the hydrothermal process. Their corresponding pseudo-first-order rate constants of the degradation were calculated and compared (Appendix A). As shown in Figure 1, the presence of raw rutile TiO_2_ shows little impact on the ozonation rate of phenol, while raw anatase TiO_2_ even has a suppressing effect on the ozonation. These results are consistent with many previous research works. Molnar et al. applied P25 TiO_2_ in ozonation and found that a TiO_2_ catalyst could not increase the efficiency of NOM removal in the ozonation process [22]. Dr. Ovelleiro’s group examined P25 and synthesized TiO_2_ for catalytic ozonation and concluded that these TiO_2_ catalysts had no effect or even a negative effect on the ozonation [23,24]. Chen et al. prepared a TiO_2_/Al_2_O_3_ catalyst and found the catalyst could not increase its removal efficiency but could enhance its TOC removal rate [25]. After the hydrothermal treatment with NaOH, the catalytic activities of TiO_2_ materials can be significantly improved, while catalysts originating from rutile show slightly lower catalytic activity than those from anatase. Among these synthesized materials, anatase treated with 15 M NaOH (A-15) shows the highest catalytic activity with a rate constant of 0.0658 ± 0.0114 min^−1^, which is 4.2 times higher than that of raw anatase TiO_2_. Compared to raw rutile TiO_2_, the catalytic rate constant of R-15 (0.0536 ± 0.0045 min^−1^) is 2 times higher. Moreover, the catalytic activity of the catalyst increases with the concentration of applied NaOH, indicating the key influence of NaOH in hydrothermal treatment during preparation. Compared to catalysts treated with 15 M NaOH (A-15 or R-15), catalysts treated with 10 M NaOH (A-10 or R-10) show slightly lower activity: the catalytic rate constants of A-10 and R-10 are 0.0586 ± 0.0067 min^−1^ and 0.0511 ± 0.0026 min^−1^, respectively. However, the washing process for catalysts treated with 15 M NaOH consumes much more acid reagent and needs more time in order to receive neutral catalysts, which is not environmentally friendly.

The catalytic rates of titania catalysts treated with 10 M NaOH without calcination (samples 110-24) are also compared with those with calcination at 400 °C (samples 110-24-400), as shown in Figure 2. The catalytic efficiencies of samples originating from anatase with or without calcination show no significant difference in ozonation of phenol, while the catalyst originating from rutile without calcination shows a slightly higher rate. Song et al. attributed the catalytic activity of TiO_2_ (including nanotubes with calcination) to its high specific surface area and its rutile crystallite phases in ozonation of phenol [7]. In addition, higher temperature or longer time of hydrothermal treatment (samples 180-24 and 110-48 in Figure 2) would depress the catalytic efficiency for ozonation of phenol, showing that the hydrothermal treatment at 110 °C for 24 h is appropriate. However, we still do not understand the mechanism of catalytic ozonation with titanate catalysts without calcination. Hence, further characterizations of these synthesized catalysts are needed in order to explore the correlation between the physico-chemical properties of the titanate materials and their catalytic performance in ozonation.

### 2.2. Physico-Chemical Characteristics of Synthesized Titania Catalysts

The synthesized materials were firstly checked with HR-TEM for their morphology and crystallinity. As shown in Figure 3, a small number of titania nanotubes can be observed in sample A-5, while more nanotubes are displayed in the samples with a higher concentration of NaOH (i.e., 7.5 M and 10 M). In samples A-5 and A-7.5, the lattice spacings of anatase are measured to be 3.55 Å and 3.56 Å, respectively, which are very close to the value of raw anatase (3.54 Å). Almost all of the materials could be transformed into nanotubes when the samples were treated with 10 M NaOH, which are several hundred nanometers to several micrometers in length with a diameter of 5 to 40 nanometers (internal diameter of several nanometers). Regarding these samples (treated with NaOH ranging from 5 M to 10 M), the treatment with higher concentrations of NaOH could bring more nanotubes into the sample as well as show higher catalytic activity. The higher proportion of nanotubes in the sample may be one reason for the higher catalytic activity. When the anatase was hydrothermally treated with 15 M NaOH, the sample exhibited the highest catalytic activity; however, the nanotubular structure of this sample was completely destroyed (Figure 3g–h). Similar observations can be obtained with respect to the rutile samples treated with different concentrations of NaOH (see Appendix A). Hence, we found that the nanotubular morphology is not mainly responsible for the high catalytic efficiency of the synthesized samples, which is consistent with previous work [7].

XRD analysis was further applied for the phase composition and crystal information of these catalysts, the results of which are exhibited in Figure 4. When comparing the XRD patterns of the synthesized catalysts with the original TiO_2_ material (raw anatase or rutile), the samples that underwent NaOH hydrothermal treatment tended to change their crystal characteristic diffraction peaks: higher NaOH concentration would bring more change in XRD diffraction. Many works have stated that nanotubes originating from TiO_2_ are actually titanate: sodium titanate and/or hydrogen titanate [26,27,28,29]. According to JCPDS card No. 47-0124, the catalysts A-10 or R-10 are mainly identified as H_2_Ti_2_O_5_∙H_2_O, which is consistent with previous works [19,27,30]. In these two samples, the diffraction peak of anatase or rutile TiO_2_ would be minimized. Hence, the anatase and rutile could be transformed into titanate as the NaOH concentration increased in the range of 0–10 M. In addition, the anatase samples that underwent NaOH hydrothermal treatment show less sharpness in their characteristic peaks and more amorphous structures, indicating lower crystallinity and smaller crystal size. The average crystal sizes and lattice spacings of anatase D(101) or rutile D(110) of these samples were calculated based on the Scherrer formula and Bragg’s Law [31,32,33] (see Table 1), while the anatase (101) peak of A-10 and A-15 as well as the rutile (110) peak of R-15 are absent in their XRD patterns, which is consistent with the data from TEM analysis. Nevertheless, the titanate diffraction peaks cannot be recognized in samples A-15 and R-15, most probably because they have been transformed into amorphous titanate due to the overdosed NaOH. Furthermore, materials originating from anatase or rutile TiO_2_ show similar transformation trends when changing the NaOH concentration in the hydrothermal process.

The BET surface area and pore size distribution of synthesized titania materials were measured (shown in Table 1, Figure 5 and Appendix A). Generally, the surface area of titania material, no matter whether originating from anatase or rutile, increased with increases in the applied concentration of NaOH in the hydrothermal process due to the formation of more nanotubes. TiO_2_ samples treated with 10 M NaOH, A-10 and R-10, showed the highest BET surface areas of 219.3 m^2^/g and 228.8 m^2^/g, respectively. When overdosed NaOH was applied (i.e., 15 M), the surface area of the material reduced sharply because of the wrecked nanostructure. Hence, a higher specific surface area may not have a direct correlation with the higher catalytic efficiency of the synthesized catalysts. In addition, these titania materials show a relatively narrow distribution in pore size based on the Barrett–Joyner–Halebda (BJH) desorption data: around 3.6 nm and 8.0 nm for A-10, and around 3.4 nm and 22 nm for R-10. The pore sizes of 3.6 nm and 3.4 nm are inferred to be the internal diameters of the nanotubes, which agree well with the observations of TEM analysis.

The materials were further characterized with Raman spectrometry, the results of which are exhibited in Figure 6. Five Raman bands at around 143, 197, 395, 515 and 638 cm^−1^ can be identified in the spectrum of the raw anatase TiO_2_, which is consistent with the literature reports [31,34]. After hydrothermal treatment with NaOH (5 M–10 M), new Raman bands of 192, 275, 449, 667, 696, 835 and 920 cm^−1^ can be recognized, which correspond to the titanate nanotubes [26,30,35]. With respect to A-15, its Raman spectrum shows very notable features, containing several strong and sharp peaks at the lower frequency bands ranging from 80 to 310 cm^−1^. According to the research work of Dr. Bamberger et al. [36], titanate (Na_2_Ti_6_O_13_ or H_2_Ti_6_O_13_) is considered to be the main species of this synthesized material. On the basis of the above observations, the sample prepared with a higher concentration of NaOH (5–15 M) formed more titanate (Ti_2_O_5_^2−^ or Ti_6_O_13_^2^^−^), which hints at a higher amount of titanate in the sample. This may be one reason for the higher catalytic efficiency of the synthesized samples.

Many research works have argued that the surface OH groups of the catalysts play an important role in catalytic ozonation [7,37,38,39]. Hence, the densities of the surface hydroxyl groups of the synthesized materials were evaluated through FTIR analysis and a saturated (de)protonation approach according to the previous literature [7]. The vibration at around 1630 cm^−1^ in the FTIR spectra (Figure 7c,d) can be attributed to O-H bending vibration, while that at 3400 cm^−1^ corresponds to the OH stretching vibration [40]. Moreover, the intensity of vibrations increases with the concentration of NaOH used in hydrothermal treatment, indicating that a higher concentration of NaOH would bring more OH groups onto the surface of the titania samples. The results of saturated (de)protonation also show that the density of surface OH groups of catalysts generally increases with the concentration of NaOH applied during the hydrothermal process (see Figure 7a,b). Among the synthesized catalysts, samples A-15 and R-15, which show the highest catalytic activities, have the highest density of surface OH groups of 1.43 mmol/g and 1.51 mmol/g, respectively. Furthermore, the catalytic rate constant is found to be nearly proportional to the density of surface hydroxyl groups of catalyst, so that a linear relation can be recognized, as shown in Figure 7a,b. Therefore, a high density of surface OH groups of catalyst should be essentially responsible for the high catalytic efficiency of the titania catalysts. Furthermore, the material originating from anatase shows a slightly higher density of surface OH groups compared with that from rutile, which hints at the role of raw materials.

On the basis of the above discussion, a high density of surface OH groups of a catalyst, instead of nanotubular morphology or high surface area, should be the essential reason for the high catalytic activity of the synthesized titania catalysts. According to previous works, the synthesized titanate materials are rich in surface OH groups, especially in aqueous environments [41,42,43,44]. Hence, the synthesized materials with more titanate would have higher amount of hydroxyl groups, which would eventually result in higher catalytic activity. Furthermore, the catalysts prepared from anatase show slightly higher activity than those from rutile, hinting at the minor impact of raw materials.

### 2.3. Active Oxidizing Species of Synthesized Titanate Catalysts

Many active oxidation species, such as hydroxyl radicals (^•^OH), singlet oxygen (^1^O_2_) and superoxide radicals (^•^O_2_^−^), can be detected in the catalytic ozonation process [45,46,47]. In this work, EPR spectroscopy was employed in order to determine the main active oxidizing species and their roles in the catalytic ozonation (see Figure 8 and Appendix A).

As shown in Figure 8a, clear characteristic peaks of DMPO-^•^OH with hyperfine splitting constants of $\alpha_{N}=\alpha_{H}=14.9\;G$ and a height ratio of quartet lines of 1:2:2:1 can be observed in all systems with or without catalysts, indicating the existence of ^•^OH during the catalytic ozonation or simple ozonation in this work. The peak intensity of the system with catalyst A-10 is slightly lower than that without the catalyst, suggesting ^•^OH is not mainly responsible for catalytic ozonation in this work. Although some research works have argued the significant importance of ^•^OH in catalytic ozonation [46,47,48,49], some others have confirmed the absence of ^•^OH in their processes [45,50]. The system with A-15 would have slightly higher intensity of DMPO-^•^OH than others, hinting at the minor role of ^•^OH in this system, probably due to the amorphous state and the extremely high density of surface OH groups of A-15.

With respect to the role of singlet oxygen in catalytic ozonation in this work, the characteristic TEMP-^1^O_2_ peaks of triplet-line spectra with equal intensities can be observed in all systems, as exhibited in Figure 8b. Moreover, the intensities of the characteristic peak of ^1^O_2_ would be enhanced when synthesized titanate catalysts were applied in the system. Hence, ^1^O_2_ is considered to be a dominant reactive oxidizing species that can accelerate formation in the catalytic process, which contributes to the higher catalytic efficiency of titanate catalysts. Similar phenomena are observed in Figure 8c of the DMPO-OOH EPR spectra, which suggests the synthesized catalysts can improve the generation of ^•^O_2_^−^ in the system [51]. The EPR spectra of material originating from rutile (R-10) show very similar characteristics to those from anatase. Therefore, the singlet oxygen and superoxide radicals are the main active oxidation species of which more can be produced in the catalytic process, while the hydroxyl radical may only play a minor role (as shown in Figure 1).

## 3. Materials and Methods

### 3.1. Materials

The chemicals, including anatase titanium dioxide (99.8%, 25 nm), rutile titanium dioxide (99.8%, 25 nm), sodium hydroxide (NaOH, 96%), and phenol (99~100%) were purchased from Macklin Biochemical Co., Ltd. (Shanghai, China). Hydrochloric acid (HCl, 37%) was purchased from Lingfeng Chemical Reagent Co., Ltd. (Shanghai, China). Anhydrous ethanol (≥95%) was purchased from Anhui Ante Food Co., Ltd., while Acetonitrile (99.9%) was purchased from Aladdin Biochemical Technology Co., Ltd. (Shanghai, China). All chemicals were applied directly without further purification.

### 3.2. Synthesis of TiO_2_ Nanomaterials

The synthesis of TiO_2_ nanotubes was modified based on the hydrothermal approach reported previously [21]. The general procedure of synthesis is described below: 3 g of titanium dioxide (anatase or rutile) was dispersed in 75 mL of NaOH solution of different concentrations (5 M, 7.5 M, 10 M, and 15 M) in a Teflon-lined autoclave and thermally treated at 110 °C for 24 h. After cooling down, the obtained product was washed with 0.1 M HCl until the pH of supernatant was around 3, and washed with deionized water until the pH went to neutral. Finally, the solid product was dried with a freeze-dryer (LC-10N-50, Shanghai Lichen-BX Instrument Techonology Co., Ltd., Shanghai, China) to produce the as-synthesized TiO_2_ nanomaterials. The synthesized samples originating from anatase were denoted as A-5, A-7.5, A-10, and A-15, while those originating from rutile were denoted as R-5, R-7.5, R-10, and R-15.

### 3.3. Experiments of Catalytic Ozonation

In order to investigate the catalytic activities of different TiO_2_ nanotubes in ozonation, the experiments for the destruction of phenol (initial concentration: 20 mg/L) were conducted in a self-installed device, including an ozone generator (3S-T, Beijin Tonglin Techonology Co., Ltd., Beijing, China), a mass flow controller (D07-19B, Beijin Sevenstar Flow Co., Ltd., Beijing, China), an ozone concentration analyzer (3S-J5000, Beijin Tonglin Techonology Co., Ltd., Beijing, China) and a glass reactor (120 mm (ø) × 150 mm (h)) with a thermal bath (22 ± 3 °C), as shown in Appendix A. Generally, 1 g of catalyst was dispersed in 1 L of reaction solution in the reactor with continuous stirring; the O_3_ and air mixture was introduced into the reactor at a flow rate of 50 mL/min; the concentration of O_3_ was adjusted to be 10 mg/L; 3 mL liquid samples were taken after certain time intervals and filtered with PTFE syringe filters (0.45 µm). The residual O_3_ in the off-gas was absorbed by KI solution.

The concentration of phenol in the obtained samples was quantified using a high-performance liquid chromatograph (HPLC, e2695, Waters, Milford, USA) with a 2489 UV/VIS detector set at 270 nm. An XBridge C_18_ column (250 mm × 4.6 mm, 5 µm particle size) was employed as the stationary phase, while a mixture of water and acetonitrile in a ratio of 50:50 (*v*:*v*) was applied as the mobile phase. The column temperature was set to be 30 °C, while the flow rate and injection volumes were 1 mL/min and 20 µL, respectively.

### 3.4. Characterization of TiO_2_ Nanomaterials

The properties of the TiO_2_ nanomaterials were characterized by several techniques. The XRD analysis was carried out on an Ultima IV multipurpose X-ray diffraction system (Rigaku, Tokyo, Japan) with Cu Kα radiation at 40 kV and 40 mA (λ = 1.5406 Å). JADE 6.5 software was applied for XRD spectra analysis. The Raman measurements were performed on a HORIBA JY LabRAM HR Evolution Raman spectrometer (HORIBA Scientific, Paris, France). The porosity as well as BET surface area of TiO_2_ nanomaterials was investigated by applying a Micromeritics ASAP 2020 Surface Area and Porosity Analyzer (Micromeritics, Norcross, United States). The HR-TEM images were obtained using a FEI Talos F200S Transmission Electron Microscope (ThermoFisher, Waltham, USA) with a field emission-transmission gun at 200 kV, using a Digital Micrograph for image analysis. Fourier Transformed Infrared (FTIR) spectra in the transmission mode were measured with a Thermo Scientific Nicolet iS20 FTIR spectrometer (ThermoFisher, Waltham, USA) using KBr pellets. The electron paramagnetic resonance (EPR) experiments were carried out by employing EMXPlus in situ EPR spectrometer (Bruker, Karlsruhe, Germany) with DMPO or TEMP as trapping agents.

## 4. Conclusions

In the present work, titania catalysts synthesized with simple NaOH hydrothermal treatment without calcination were applied in ozonation for phenol removal. The catalysts treated with higher concentrations of NaOH contained more titanate, which facilitated more surface OH groups on the catalyst surface and eventually resulted in better catalytic activity. Furthermore, the main active oxidizing species of catalytic ozonation were recognized as singlet oxygen and superoxide radicals.

## Data Availability

All data is available in the main text or the Appendix A.

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
