# Peer review of "Mechanism of One-Step Hydrothermally Synthesized Titanate Catalysts for Ozonation"

_molecules, 2022, doi:10.3390/molecules27092706_

Round 1

Reviewer 1 Report

This study explored the TiO2 (titanate) nanotube catalyst for ozonation was synthesized with simple one-step NaOH hydrothermal treatment without energy consuming calcination. As I can see, the MS is well organized and written. The content of this study sounds scientifically interesting. I recommend the MS to be published in Molecules after major revision. Several specific reviewing comments are presented as follows,

  1. Is the order of Results and Discussion and 3. Materials and Methods reversed?
  2. Why choose phenol as the target rather than other organic pollutants? Please explain it in the introduction.
  3. What is the basis for the phenol concentrations?
  4. Please provide more detail information about the LHSV calculations used in this study.
  5. In Figure S1, the name of samples needs to be checked, (e), (f), and (g), (h) seem to be R-10 and R-15, respectively.
  6. Line 165, the name of Table 1 needs to be modified.
  7. In Table 1, how were the crystal size and lattice spacing NA? please explain in the manuscript.
  8. In Table 1, sample A-15 and R-15 surface area were low than the A-10 and R-10, because of the wrecked structure, why? Have tested anatase hydrothermally treated with 12.5 M (take the intermediate value of 10M and 15M) NaOH?
  9. Line 209, How is “the density of surface OH groups of the catalysts” obtained? Please make a detailed description.
  10. Does EPR analysis of rutile titanate nanotube have the same pattern as anatase titanate nanotube? Please supplement the result of EPR.
  11. The MS lacks relevant references published in Molecules. Besides, adding more recently published references would increase the readability from Molecules readers for this research work.

Reviewer 2 Report

The article "Mechanism of one-step hydrothermally synthesized titanate catalysts for ozonation" presents the comprehensive and detailed study of titanate nanotube catalyst for ozonation. The TiO2 nanotubes have been obtained from simple one-step NaOH hydrothermal treatment without energy consuming calcination. A wide scope of physico-chemical methods of analysis have been employed (e.g. XRD, FTIR/Raman spectroscopy, TEM, EPR). The catalytic activity of the obtained samples has also been tested. On the basis of the obtained information on the correlation between the properties of titania and the catalytic performance the authors suggested an important role of OH groups density on the catalyst surface.

The present work is a solid research that can be published in Molecules. Nevertheless, a few following comments should be considered by the authors:

1) Please, do not hesitate to proofread for English (e.g. line 25 I guess "which limited" should be replaced by "which limits").

2) Figure 1 should be colored instead of black&white. In present version it is difficult to read.

3) The authors compare the average crystal sizes obtained from XRD with those from TEM. Did they perform a statistically valid analysis of the crystallite size distribution from TEM data? If yes, please provide the corresponding histograms in SM.

4) Tacking in account the importance of the information obtained from FTIR spectra the authors may consider a possibility to move Fig.S3 from SM to manuscript and unite it with Fig.6, for example.

Reviewer 3 Report

Journal                        Molecules

Manuscript ID: molecules-1677666
Type of manuscript: Article
Title: Mechanism of one-step hydrothermally synthesized titanate catalysts
for ozonation
Authors: Geshan Zhang *, Anhua Jiang, Xinwen Huang *, Tian Yuan, Hanrui Wu,
Lichun Li, Zongjian Liu

To increase the activity of titanate in the processes of water and wastewater treatment with ozone during the hydrothermal treatment of titanium dioxide with a NaOH solution, a catalyst for titanate nanotubes was synthesized. The catalyst is more catalytically active in removing phenol due to the higher content of OH groups on its surface. It is assumed that the main active oxidizing agents during ozonization are singlet oxygen and superoxide radical. New interesting data on the structural and electronic properties of catalysts are presented. The work is of interest to specialists in the field of technology for obtaining efficient catalysts based on titanium dioxide and further development of the ozonation mechanism with their participation. Of particular interest are the results on the study of ozonation intermediates by the EPR method.

The results of the study meet the requirements of the journal and are useful for specialists.

The work can be recommended for publication, taking into account the following remarks.

  1. Ls 74-76: Fig. 1 shows changes in phenol concentration during catalytic ozonation with TiO2 from anatase or rutile. “…Their corresponding pseudo-first order rate constants of the degradation are calculated and compared….”. Really the first order of the reaction is not shown. It is necessary to present the data of Fig. 1 in the coordinates lnC0/C versus t. The slope of the lines makes it possible to calculate the rate constants, the values of which are compared later in the text.
  2. Figure 1. It is necessary to give the parameters of the catalytic process. The concentrations of reagents at which the reaction corresponds to the first order should be indicated in the text.
  3. Lines 89-91. The article gives constants, for example, 0.0658 min-1 and 0.0536 min-1. How much is this? I think it's better to specify them in seconds and with an error
  4. Line 118: “Physiochemical characteristics”. It will probably be right “Physico-chemical characteristics”.
  5. It is desirable to give errors of the parameters given in Table 1.
  6. Lines 217-218: “Figure 7. The linear relation between catalytic rate constant and density of surface hydroxyl groups of catalyst originating from anatase (a) and rutile (b) TiO2”.

What does the fact that the lines in the figures do not come out of zero mean? What does the segment cut off on the “catalytic rate constant” axis mean when the value of “density of surface hydroxyl groups of catalyst” is zero?

In general, the work is useful and performed at a good level. It can be recommended for publication with consideration of comments.

Round 2

Reviewer 1 Report

This study explored the TiO2 (titanate) nanotube catalyst for ozonation was synthesized with simple one-step NaOH hydrothermal treatment without energy consuming calcination. As I can see, the MS is well organized and written. The content of this study sounds scientifically interesting. I recommend the MS to be published in Molecules after major revision. It can be accepted to publish now.
